# Gender inequity in speaking opportunities at the American Geophysical Union Fall Meeting

Heather L. Ford [1], Cameron Brick[2], Karine Blaufuss[3] & Petra S. Dekens[4]

Implicit and explicit biases impede the participation of women in science, technology, engineering, and mathematic (STEM) fields. Across career stages, attending conferences and presenting research are ways to spread scientific results, find job opportunities, and gain awards. Here, we present an analysis by gender of the American Geophysical Union Fall Meeting speaking opportunities from 2014 to 2016. We find that women were invited and assigned oral presentations less often than men. However, when we control for career stage, we see similar rates between women and men and women sometimes outperform men. At the same time, women elect for poster presentations more than men. Male primary conveners allocate invited abstracts and oral presentations to women less often and below the proportion of women authors. These results highlight the need to provide equal opportunity to women in speaking roles at scientific conferences as part of the overall effort to advance women in STEM.

[1] Department of Earth Sciences, University of Cambridge, Cambridge CB2 3EQ, UK. [2] Department of Psychology, University of Cambridge, Cambridge CB2 3EB, UK. [3] American Geophysical Union, Washington 20009 District of Columbia, USA. [4] Department of Earth & Climate Sciences, San Francisco State University, San Francisco 94132 California, USA. Correspondence and requests for materials should be addressed to H.L.F. (email: hlf40@cam.ac.uk)

There are conscious efforts underway to increase gender equity in STEM fields such as the National Science Foundation's ADVANCE: Increasing the Participation and Advancement of Women in Academic Science and Engineering Careers program. However, despite numerous initiatives to increase the enrollment and retention of women in STEM, the causes of the continued gender disparity remain unclear[1]. Evidence suggests that implicit and explicit biases hinder the participation of women in STEM fields[2]. Many gender-related biases are documented from disparities in the strength of letters of recommendation[3], solicitation to review research articles[4], and academic pay[5].

Attending and presenting at conferences is one way researchers expand their network, seek collaborators, connect with mentors, and improve research visibility. In particular, presenting research as an invited speaker and giving an oral presentation are ways to efficiently disseminate scientific results and build one's career. Speaking at a conference is important for career advancement across career stages, particularly for finding job opportunities and funding, and gaining awards and recognition.

The American Geophysical Union (AGU) Fall Meeting is the world's largest geoscience conference with over 22,000 abstract submissions each year. The meeting covers a wide breadth of Earth and space sciences such as atmospheric sciences, volcanology, and space physics. The AGU Fall Meeting provides a high-powered test for equality in the allocation of speaking opportunities to men and women across a broad range of physical sciences.

Since 2013, AGU, an international scientific association with 60,000 members from 137 countries, has asked its members to self-report gender (female, male, prefer not to answer), highest degree obtained, including year, and other demographic data. For the AGU Fall Meeting 2014 to 2016 abstract database (here after referred to as the abstract database), 98% ($n = 65,247$) of abstract authors self-identify as male or female, of which >98% provided career information ($n = 64,209$). Note that although authors self-identify their gender, our binary analysis (female/women/male/men) does not capture the spectrum of gender identity.

Career stage is self-identified as student or retired, or calculated based on number of years since highest degree obtained: early career (0–10 years), mid-career (10–20 years), experienced (>20 years). AGU defines these career stages for award eligibility.

AGU is organized into Sections (e.g., atmospheric sciences, seismology, etc). For the AGU Fall Meeting, topical sessions within a section are proposed by a self-organized group of up to four members, led by a primary convener. Traditionally, there are two types of sessions: oral and poster. The primary convener and the co-convener(s) can also invite a limited number of authors.

During abstract submission, authors opt to be assigned an oral or poster presentation by the conveners or may opt for a poster only presentation. The primary convener and co-convener(s) then assign abstract submissions as either oral or poster presentation. When an author opts for a poster only abstract submission, it typically remains a poster presentation (99%). The abstract first author and primary convener must be AGU members; however, this constraint does not apply to invited authors, co-author(s), and co-convener(s). Most invited authors are AGU members. The abstract database does not currently include gender and career stage information for co-author(s) and co-convener(s). Therefore, we do not test for possible co-author(s) and co-convener(s) influence on gender parity.

The AGU membership is representative of those actively engaged in academic, government and industry research within the United States[4]. Women comprise 28% of the AGU membership, which is similar to the percentage of women currently employed in physical sciences (chemists and material scientists: 30%; environmental scientists and geoscientists: 24%; other physical scientists 38%)[6] and science and engineering occupations (28%)[7].

The opportunity to speak is fundamental to career advancement across career stages for job opportunities, collaborations, awards and recognition. Here, we find that female scientists are offered fewer speaking opportunities than men overall. However, these results are influenced by the gender demographics of AGU where women disproportionally occupy the student career stage and few speaking opportunities are offered to students. When we control for career stage, we see similar rates of oral presentations between women and men, and women at early career and mid-career stages are invited authors more often than men. Across career stages, primary convener men, who control 72% of the abstract pool, provided fewer opportunities to women. Promoting student and early career women speakers and women acting in primary convener roles will improve overall gender parity in scientific conferences.

## Results

**Gender demographics of the AGU Fall Meeting.** Women submitted 32% of all abstracts ($n = 20,900$) and are concentrated in the student and early career stages (77% of women vs. 60% of men, Fig. 1). This distribution of women reflects the leaky pipeline and the historical barriers for participation for women in STEM fields[8, 9].

**Speaking at conferences.** The chi-squared test ($\chi^2$) is used throughout to test for differences between expected and observed frequencies and the specific hypotheses tested are listed in the "Methods" section. Overall, fewer women than men are given the opportunity to highlight their research through invited abstracts and oral presentations (Fig. 2). However, this result is impacted by the gender distribution of AGU. The most common career category for women is student (39% of women authors are students vs. 25% of men) and students have fewer speaking opportunities overall (i.e., students comprise 4.8% of invited abstracts and 15% of oral presentations).

Overall, women were invited to submit abstracts at a lower rate than men [10 vs. 12%, Figure 2a, $\chi^2(1, 65246) = 96.8$, $p < 0.001$, Supplementary Table 1]. Of invited authors ($n = 7539$), 31% were

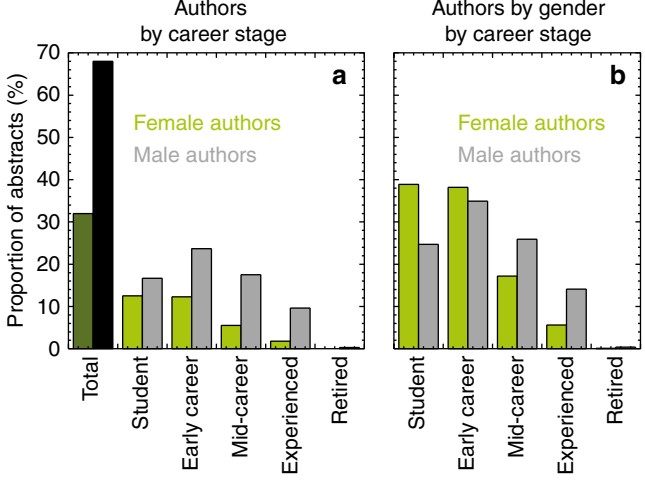

**Fig. 1** American Geophysical Union Fall Meeting Gender Demographics. Proportion of total abstracts by career stage (**a**) shows male authors are a large portion of submitted abstracts. Proportion of abstracts by gender by career stage (**b**) shows female authors are concentrated in the student and early career stages

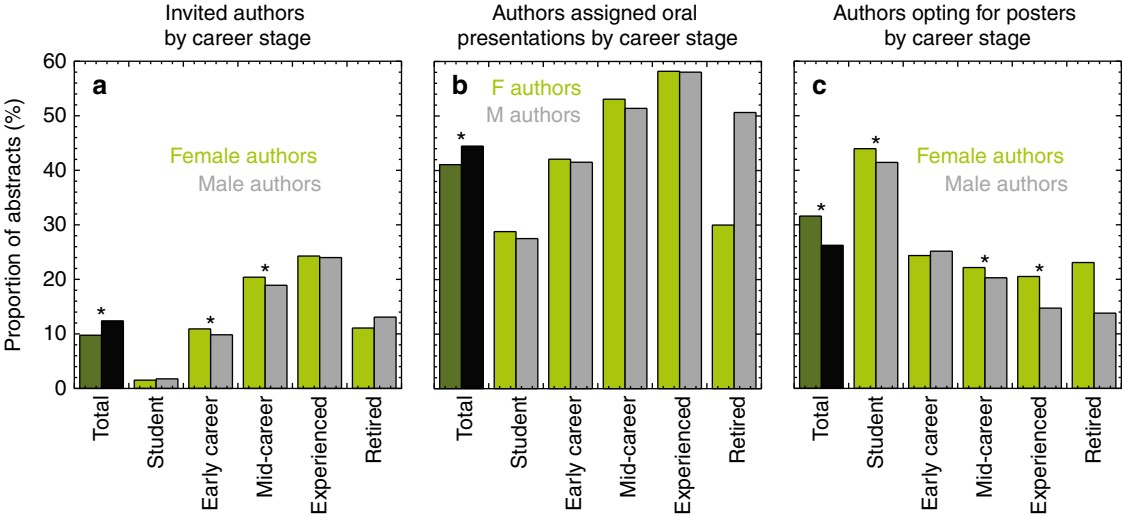

**Fig. 2** Author submissions to American Geophysical Fall Meeting by gender and career stage. Proportion of invited authors (**a**) shows overall men are invited at a higher rate than women; however, when controlling for career stage, early career and mid-career women are invited at a higher rate than male colleagues in their cohort. Authors assigned oral presentations (**b**) shows overall men are assigned oral presentations at a higher rate than women, but when controlling for career stage there is no significant difference. Authors opting for posters (**c**) shows women often opt for poster presentations more often than men, both overall and at some equivalent career stages. Totals shown here are the proportion of total abstracts. An asterisk indicates a significant result at $p < 0.05$

early career ($n = 2363$) and 38% were mid-career ($n = 2859$). Women are invited at a significantly higher rate than men within the early career (10.9 vs. 9.9%, $\chi^2(1, 23111) = 6.18$, $p = 0.013$) and mid-career (20.4 vs. 18.9%, $\chi^2(1, 14814) = 3.866$, $p = 0.049$) stages. The early career stage includes postdoctoral training, which for women is the leakiest part of the STEM career pipeline[10], due in part because these are typically family formation years[1]. Women are also more likely than men to spend more time in postdoctoral positions before securing tenure-track jobs[11]. AGU states the objective of invited authors are to raise the profile of the session and to attract "authors who would not otherwise submit an abstract to a session in an effort to, for example, enhance diversity or interdisciplinary perspectives or feature early-career scientists."

For logistical reasons, in 2016 the AGU reduced the number of invited abstracts a primary convener could invite from four to two. Notably, this change was associated with a reduction in the gender bias for invited abstracts. That is, although women continued to be invited to submit abstracts at a lower overall rate than men, the difference between women and men was less in 2016 than 2014/2015 [2014/2015: $\chi^2(43,535) = 81.0$, $p < 0.001$; 2016: $\chi^2(21,710) = 14.1$, $p < 0.001$; difference: $\chi^2(1) = 66.9$, $p < .001$].

Of all authors that opt to be assigned to an oral or poster presentation by the conveners ($n = 31,348$), women were assigned oral presentations at a lower rate than men [41.1 vs. 44.5%, Figure 2b $\chi^2(1, 31347) = 31.1$, $p < 0.001$, Supplementary Table 2]. When controlling for career stage, there is no significant difference between women and men.

Although some Sections within AGU have a larger proportion of women (e.g. Public Affairs, now incorporated into Societal Impacts and Science Policy), there is no significant relationship between the proportion of women and the rate of invited abstracts [$r(23) = 0.02$, $p = 0.92$] and oral presentations [$r(22) = 0.12$, $p = 0.57$] between men and women (Supplementary Figure 1, Supplementary Table 3).

**The role of the primary convener.** The primary convener leads the decision to invite and assign oral or poster presentations for a specific session. Male and female primary conveners invited women authors 24% ($n = 1302$) and 34% ($n = 716$) of the time, respectively (Figure 3a, Supplementary Table 4). Men primary conveners invited fewer women authors at early career, mid-career, and experienced career stages. Male and female primary conveners assigned women authors oral presentations 29% ($n = 3769$) and 37% ($n = 1733$) of the time, respectively (Figure 3c, Supplementary Table 5). Men primary conveners assigned fewer women authors oral presentations at student, early career, mid-career, and experienced career stages.

We also examined whether there were differences in inviting and assigning oral presentations by career stage of the primary conveners themselves. From student to more senior career stages, men primary conveners invited (Fig. 3b, Supplementary Table 6) and assigned (Fig. 3d, Supplementary Table 7) fewer women than women primary conveners, and below the proportion of women available in the abstract pool. Thus, regardless of primary convener career stage, primary convener men provided fewer opportunities to women.

**Women opt out.** Women elect for poster only presentations more than men [32 vs. 26%, Figure 2c, $\chi^2(1, 43514) = 134.9$, $p < 0.001$, Supplementary Table 8]. This relationship is significant across the student (44 vs. 41%), mid-career (22 vs. 20%), and experienced (20 vs. 15%) career stages.

## Discussion

Attending conferences and interacting with colleagues is vital to the exchange of ideas within the science community. By giving oral presentations, scientists increase professional visibility, widely disseminate results and improve their communication skills. Recent research shows that extending a larger portion of invited and oral presentations to first-time presenters (i.e., student and early career stages) improves the overall parity in speaking opportunities[12]. Similarly, our results show choosing student and early career stages for invited abstracts and oral presentations may help, as women are concentrated in these career stages. Ninety-three percent of invited abstracts and 83% oral presentations are allocated to more senior career stages

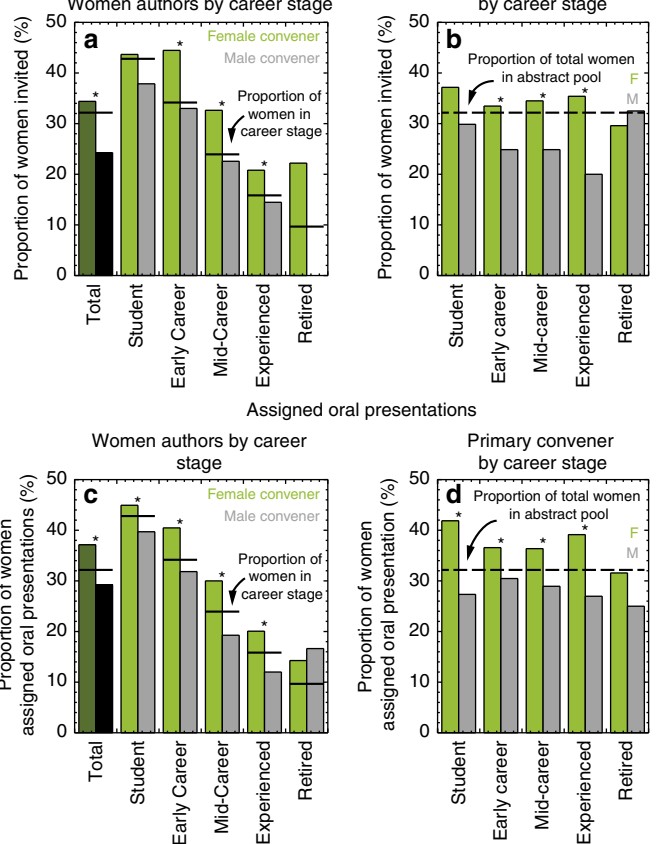

**Fig. 3** Primary convener allocations for American Geophysical Fall Meeting. Proportion of women across career stages invited by primary convener gender (**a**) shows men primary conveners invite fewer women and the proportion of women invited by primary conveners' gender and career stage (**b**) shows this is true across men primary conveners' career stages. Proportion of women across career stages assigned oral presentations by primary convener gender (**c**) shows men primary conveners assign oral presentations to fewer women and the proportion of women assigned oral presentations by primary conveners' gender and career stage (**d**) shows this is true across men primary conveners' career stages. An asterisk indicates a significant result at $p < 0.05$

where there are fewer women due to the leaky pipeline and the historical barriers women face in many STEM fields. AGU and other scientific conference organizers should encourage the science community to promote junior attendees in speaking roles.

Because men primary conveners control a larger portion of abstracts [72% of all abstracts ($n = 47,812$)], their higher preference for other men compared to female primary conveners has a disproportionate impact on the visibility of women as invited authors or oral presentation speakers. Male conveners offered fewer invited abstracts and speaking opportunities to women; this implies the reason AGU has gender parity when controlling for career stage is because women disproportionally invite other women. This suggests the underrepresented gender is doing the burden of gender parity efforts. The presence of women on speaker selection committees improved parity in virology conferences[12] and colloquium speakers at top universities[13]. AGU should encourage women to act as primary conveners. Women co-conveners may also help improve parity.

The confidence gap[14] may explain why women disproportionally opt for poster presentations. Women tend to

underestimate their ability and performance in science[15]. Electing to present a poster may be more desirable if women feel their research is not ready for an oral presentation, and/or an oral presentation feels like a high-stakes performance. Alternatively, women may opt for poster only presentations because presentation times are more flexible and/or they feel posters might provide more networking opportunities.

There is a myriad of reasons why women may leave academic STEM fields. Implicit biases—from the quality of recommendation letters[3], funding[16] speaking opportunities[12, 13], and recognition and medals[17]—and explicit biases, such as structural biases that impede family formation and parental support[1], hinder the progression of women into more senior roles and their research visibility. Reducing gender bias in speaking roles is critical for the advancement of women in science. Encouraging more women to act as primary conveners and colloquium organizers may also reduce the overall gender imbalance[12, 13], so AGU and other scientific conference organizers should promote women in these roles. All conveners may benefit from implicit bias training prior to inviting and assigning oral presentations to speakers, although additional research is needed on training efficacy[18, 19].

The AGU membership and abstract database is evolving into a rich resource to explore diversity and equity. AGU started collecting demographic data, including gender and ethnicity (members affiliated with institutions in the United States of America) from its membership in 2013. As response rates increase, opportunities for nuanced demographic analyses will become possible. It is particularly important to investigate the data as it pertains to underrepresented minority groups because geoscience is one of the least ethnically diverse STEM fields[20]. AGU is in the process of importing all of the abstract and membership data so that co-convener demographic information will also be available for future analyses.

Small interventions may improve the visibility of women in science. For instance, after an analysis of gender bias in peer review within AGU publications[4], AGU now includes a statement asking authors to help improve the diversity of the reviewer pool during the manuscript submission process[21]. This small intervention improved the gender diversity of the suggested reviewer pool, particularly for male authors. We recommend AGU and similar scientific organizations include statements asking primary conveners and conference organizers to be mindful of diversity. Mindfulness about diversity issues, particularly among men, while making decisions on speaking opportunities, reviewer suggestions, job candidates, and scientific recognition in medal and awards will improve community diversity.

## Methods

**The database**. The American Geophysical Union (AGU) organizes the largest physical sciences meeting internationally with over 22,000 abstracts submitted each year in our 2014–2016 database. Since 2013, AGU has asked its members to self-report demographic information including gender, highest earned degree, and year in which highest degree was earned. AGU membership is required to submit an abstract and to act as a primary convener of a session.

The AGU organizes sessions within Sections. According to AGU, these Sections "are responsible for fostering scientific discussion and collaboration among members who affiliate with them." The primary convener and co-convener(s) submit a session proposal to a particular Section in April. A session proposal is self-organized around a scientific topic that may be of broad interest within a Section. In June, the session proposal is reviewed for approval by the Program Committee.

After approval, the primary convener and co-convener(s) may invite authors (up to four in 2014 and 2015, up to two in 2016) to submit abstracts. We call these Invited Authors. Members of the broader AGU community are able to submit abstracts to a session until the submission deadline in August. At the time of submission, authors request "Assigned by Program Committee (Oral or Poster)" or "Poster Only." The author that submits an abstract (invited or otherwise) we call the First Author.

After the submission deadline, the Program Committee determines the available number of oral and poster sessions for each Section based on the submission numbers and available space within the convention center. The Secretary within a

Section then allocates the available oral and poster sessions to each session proposal. If only a few abstracts are submitted to a proposed session, proposed sessions may merge at this time. As a session is only allowed up to four conveners, some individuals will relinquish their convening role. One primary convener will typically stay on as primary convener while the other remains as a co-convener. We are unable to investigate the potential impact this may have on our results.

Once the oral and poster sessions have been delegated within a Section, the primary convener and co-convener(s) allocate the oral and poster presentations.

For these analyses, the data was accessed in March 2017. At the time, the "Requested Format – Assigned by Program Committee (Oral or Poster) and Poster Only" were not available for 2016. Therefore, the gender analyses on oral presentation allocation is done on the 2014 and 2015 data only.

Our variables are first author gender (female, male), first author career stage (student, early career, mid-career, experienced and retired), invited (yes, no), requested format (assigned by program committee (oral or poster) and poster only), primary convener gender (female, male), primary convener career stage (student, early career, mid-career, experienced and retired).

Career stage for first author and primary convener is self-identified as student or retired, or calculated based on number of years since highest degree obtained: early career (0–10 years), mid-career (10–20 years), experienced (>20 years). Student member status is confirmed annually by a faculty member. Unfortunately, using this method to calculate career stage overlooks career breaks that members may have taken to raise families, out of medical necessity and/or a myriad of other reasons.

**Statistical tests**. We used chi-squared test ($\chi^2$) to test our hypotheses. $\chi^2$ is used throughout to determine whether there is significant difference between the expected and observed frequencies. Symbols are $\mu$ (mean), $\sigma$ (standard deviation) and $n$ (number of individuals). Results are reported as: $\chi^2$ (degrees of freedom, sample size) = the $\chi^2$ value, and the associated $p$-value.

Hypothesis 1: Women are invited to submit abstracts at a lower rate than men [$\mu$ female = 1.098, $\sigma$ = 0.297, $n$ female = 20,900; $\mu$ male = 1.124, $\sigma$ = 0.330, $n$ male = 44,347; $\chi^2$(1, 65,246) = 96.8, $p$ < 0.001, Supplementary Table 1].

Hypothesis 2: In 2016, AGU reduced the number of invited abstracts a primary convener could invite from four to two. Women are invited to submit abstracts at a lower rate than men in 2014/2015 than in 2016 [2014/15: $\mu$ female = 1.111, $\sigma$ = 0.314, $n$ female = 13,791; $\mu$ male = 1.142, $\sigma$ = 0.350, $n$ male = 29,745; $\chi^2$(1, 43,535) = 81.0, $p$ < 0.001, 2016: $\mu$ female = 1.072, $\sigma$ = 0.258, $n$ female = 7109; $\mu$ male = 1.087, $\sigma$ = 0.282, $n$ male = 14,602; $\chi^2$(1, 21,710) = 14.1, $p$ < 0.001, $\chi^2$ = 81.0–14.1 = $\chi^2$ = 66.9, $p$ < 0.001].

Hypothesis 3: Women are invited to submit abstracts at a lower rate than men at all career stages. Women are invited to present at a higher rate in the Early Career and Mid-Career stage [Supplementary Table 1].

Hypothesis 4: Women are less likely to be assigned an oral presentation than men after requesting "Assigned by Program Committee (Oral or Poster)" [$\mu$ female = 1.589, $\sigma$ = 0.492, $n$ female = 9424; $\mu$ male = 1.555, $\sigma$ = 0.485, $n$ male = 21,924; $\chi^2$(1, 31,347) = 31.1, $p$ < 0.001]. We also repeated this test by omitting the invited speakers [$\mu$ female = 1.680, $\sigma$ = 0.47, $n$ female = 7907; $\mu$ male = 1.659, $\sigma$ = 0.47, $n$ male = 17,711; $\chi^2$(1, 25,617) = 11.0, $p$ = 0.001, Supplementary Table 2].

Hypothesis 5: Women are less likely to be assigned an oral presentation than men at all career stages after requesting "Assigned by Program Committee (Oral or Poster)" There are no significant relationships at any career stage [Supplementary Table 2].

Hypothesis 6: As the proportion of women in a Sections increases, the overall gender bias in invited authors, oral presentation assignments and poster presentations requests will go down. We find no significant correlations between invited authors, oral presentation assignments, and poster presentations requests and the proportion of women in the Section [Supplementary Figure 1, Supplementary Table 3]. Note Union is excluded from oral presentation assignments and poster presentations requests because all Union sessions are oral presentations (i.e., there are no poster sessions).

Hypothesis 7: Male primary conveners invite male abstract submissions at a higher rate than female primary conveners [$\mu$ female = 0.656, $\sigma$ = 0.475, $n$ female = 2081; $\mu$ male = 0.7571, $\sigma$ = 0.429, $n$ male = 5361; $\chi^2$(1, 7441) = 77.7, $p$ < 0.001, Supplementary Table 4].

Hypothesis 8: Male primary conveners invite male abstract submissions at a higher rate than female primary conveners and this effect emerges for each First Author (FA) career stage. This is significant at for the First Author Early Career, Mid-Career and Experienced career stages [Supplementary Table 4].

Hypothesis 9: Male primary conveners assign male speakers oral presentations at a higher rate than female primary conveners. [$\mu$ female = 0.6285, $\sigma$ = 0.483, $n$ female = 4665; $\mu$ male = 0.7076, $\sigma$ = 0.458, $n$ male = 12,888; $\chi^2$(1, 17,552) = 88.5, $p$ < 0.001, Supplementary Table 5].

Hypothesis 10: Male primary conveners assign male speakers oral presentations at a higher rate than female primary conveners and this effect emerges for each first author (FA) career stage. This is significant at for the first author student, early career, mid-career, and experienced career stages [Supplementary Table 5].

Hypothesis 11: Male primary conveners invite male abstract submissions at a higher rate than female primary conveners and this effect emerges for each primary convener (PC) career stage. This is significant at for the primary convener early career, mid-career and experienced career stages [Supplementary Table 6].

Hypothesis 12: Male primary conveners assign male speakers oral presentations at a higher rate than female primary conveners and this effect emerges for each primary convener (PC) career stage. This is significant at for the primary convener student, early career, mid-career and experienced career stages [Supplementary Table 7].

Hypothesis 13: Women request poster presentations at a higher rate than men [$\mu$ female = 1.32, $\sigma$ = 0.465, $n$ female = 13784; $\mu$ male = 1.26, $\sigma$ = 0.440, $n$ male = 29731; $\chi^2$(1, 43,514) = 134.9, $p$ < 0.001, Supplementary Table 8].

Hypothesis 14: Women request poster presentations at a higher rate than men at all career stages. This is significant at the student, mid-career, and experienced stages [Supplementary Table 8].

**Data availability**. For confidentiality reasons, the AGU membership database is not publically available. The abstract database, without demographic information, is publically available at https://meetings.agu.org/abstract_db/. Summary data and statistics are included the Supplementary Materials.

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

## Acknowledgements

This work benefitted from conversations with M. Azmitia and R. Rhodes. The manuscript improved from comments by A.M. Holmes, B. Williams, B. Hanson and J. Lerback. H.L.F. is supported by the United Kingdom Natural Environmental Research Council (NERC) Independent Research Fellowship (NE/N015045/1).

## Author contributions

H.L.F. and P.S.D initiated the study. K.B. curated the data and performed preliminary analyses. H.L.F. and C.B. conducted the analyses and C.B. performed statistical tests. H.L.F. wrote the manuscript with input from C.B., P.S.D. and K.B.

## Additional information

**Competing interests:** The authors declare no competing interests.

