## [Peer Review File · Nature Communications]

Reviewers' comments:

Reviewer #1 (Remarks to the Author):

This paper addresses an important issue in the geosciences: gender representation in meeting presentations. The authors analyze data from 3 Fall AGU meetings to determine the gender representation of invited and assigned oral presentations. They found that women gave these presentations less often, but similar rates after controlling for career stage. Furthermore, women elected more often for poster presentations and male conveners gave presentations to women below the proportion of women authors.

This paper is novel, important, and very well written. It will be of interest and will influence others in the field. The paper's claims are very convincing. There are some issues, however, that would benefit from more explanation or more work.

1. It would improve the paper to include more information about the abstract database. Is it available to everyone? Or were the authors somehow able to get this information? If so, how? Did the authors collect the information themselves? If so, how? Given that the database is unlikely to be widely available, more data analysis in this paper would be useful.
2. The paper points out that their analysis was limited by binary gender information. Do they know if the AGU plans to collect more detailed gender information in the future? If not, perhaps it could be a conclusion in the paper.
3. The paper points out that the database does not include information on the gender or career stage of the co-conveners. Is there a plan to begin to collect this information? If not, why not? Given the importance of this information and the possible influence of all-male conveners on gender representation, more discussion of this point is necessary.
4. Several questions could be added to this analysis and would be of great interest to the field:
 - a. Are women authors more likely to submit abstracts to sessions with a women primary convener?
 - b. How common are all male (or all female) sessions? (Note: I attended all male session during the Fall 2017 AGU meeting, which made me want the answer to this question. The one I attended had 8/8 speakers in the oral session and 4/4 male conveners.)
 - c. Does the gender representation vary by AGU sub discipline? AGU organizes their sessions by discipline and the analysis shown here should be able to be broken down for each of those. Unless the numbers get too small? If so, this should be discussed.

Reviewer #2 (Remarks to the Author):

Review for: Gender Representation of Speaking Opportunities at the American Geophysical Union Fall Meeting

Overall, this paper has great data on a timely, relevant, and important topic. The authors have done a good job putting together all this data. The major claim of the paper is that women are not invited as often as men to speak at the AGU Fall Meeting, a result that will be of interest to others in the field, and also to the broader STEM community. These results tie in to the larger picture of factors that hinder women's advancement in STEM fields. My suggestions/comments are:

1. Rather than merely present the data as the authors have done, they should also offer a broader narrative, i.e. give the paper a voice, so to speak. They do touch on some of the other studies that explore gender bias, but in its current form this paper does not flow very smoothly with respect to a compelling narrative. For example, rather than simply mention that their study is consistent with the finding of a leak in the pipeline, the authors could say a little more about the causes of such a leak, especially within the context of the bigger picture, and how their finding (i.e. that women are

not invited as often as men to make presentations at AGU) is one piece of a larger problem of gender bias in academia. That way, this paper becomes more relevant to a broader audience than just geoscientists who attend AGU.

2. Explain the statistics and figures a little bit more – not everyone is familiar with what a chi-square statistic represents. And the figures could also do with a bit more explanation – just a couple of sentences should do.

3. Refer to a few additional studies that are relevant to this paper – for example the Hebl paper that just came out in PNAS (Dec 2017) – that shows that women were much less likely to be invited to be colloquium speakers compared to men.

<http://www.pnas.org/content/early/2017/12/12/1708414115>

4. Explicitly offer a specific set of recommendations instead of merely hinting at what could be done. For example, lines 133-136 say that men primary conveners control a larger portion of abstracts, so their preference for inviting men over women has a disproportionate impact on the visibility of women speakers. Following this, make a recommendation that AGU committees should make it a point to include more women primary conveners, and/or that male primary conveners should seek input from others (including women peers and colleagues) before issuing invitations to speakers. One could also make a case that following the data and findings in this paper, AGU should require all primary conveners to go through a bias training, and also explain why they invited the speakers that they did, and what steps if any they took to ensure broader participation to include women speakers. The authors can further buttress this point by referring to the recent PlosOne paper that shows that in the natural sciences women are more egalitarian than men (see: <https://www.ncbi.nlm.nih.gov/pmc/articles/PMC5425184/?platform=ho+otsuite>), and so putting more women in primary convener positions would broaden participation.

5. This paper has implications for shaping policy on inviting speakers for presentations – not just AGU but potentially other STEM professional societies as well. The authors should develop this aspect further instead of merely stating the data.

6. Tie in bias (implicit or otherwise) with the leaks in the pipeline in STEM fields in academia. Be it letters of recommendation, journal reviewer activity, teaching evaluations, salary differences, grant funding, etc. – these are all pieces of a larger picture, one in which bias, combined with society's perceptions of gender-normative roles, plays a very key role in limiting women's professional advancement in STEM fields.

7. The ending seems very abrupt.

8. Overall, this paper has great data. My main concerns, as evident from my points above, are that: a) the authors have not fully developed the potential policy implications of this data; and b) the paper lacks the "connective tissue" in the form of a compelling narrative that will influence the thinking in this field and make these results accessible and relevant to a broader community. However, these limitations can be fixed with some re-writing, and do not require any major changes.

Response to Reviewer Comments on “Gender Representation of Speaking Opportunities at the American Geophysical Union Fall Meeting”

Note: Reviewer comments are in normal text, *responses to reviewers are in blue italic text*, and **manuscript revisions are in bold text**.

We thank the two reviewers for their helpful feedback. We address each point below.

Reviewer #1 (Remarks to the Author):

This paper addresses an important issue in the geosciences: gender representation in meeting presentations. The authors analyze data from 3 Fall AGU meetings to determine the gender representation of invited and assigned oral presentations. They found that women gave these presentations less often, but similar rates after controlling for career stage. Furthermore, women elected more often for poster presentations and male conveners gave presentations to women below the proportion of women authors.

This paper is novel, important, and very well written. It will be of interest and will influence others in the field. The paper’s claims are very convincing. There are some issues, however, that would benefit from more explanation or more work.

1. It would improve the paper to include more information about the abstract database. Is it available to everyone? Or were the authors somehow able to get this information? If so, how? Did the authors collect the information themselves? If so, how? Given that the database is unlikely to be widely available, more data analysis in this paper would be useful.

The database was made available by AGU by request in order to research the relationship between gender and speaking opportunities at the AGU Fall Meeting. The abstract database has a connecting point to the membership database. The demographic data (gender and career stage) was extracted from the membership database and correlated it to the abstract database. For confidentiality reasons, the membership database is not publically available. The content of the abstract database is publically available on the AGU website. However, the abstract database does not include gender, career stage or age.

The following text was **revised** in the Supplementary Materials: **For confidentiality reasons, the AGU membership database is not publically available. The abstract database, without demographic information, is publically available at https://meetings.agu.org/abstract_db/. Summary data and statistics are included in the tables below.**

2. The paper points out that their analysis was limited by binary gender information. Do they know if the AGU plans to collect more detailed gender information in the future? If not, perhaps it could be a conclusion in the paper.

For the gender data collection, AGU members have the option to choose 1) male (67.0%), 2) female (31.6%), and 3) prefer not to answer (<0.8%). We included the statement “Note that although authors self-identify their sex, our binary analysis (female/women/male/men) does not capture the spectrum of gender identity” to acknowledge that members may identify differently

Response to Reviewer Comments on “Gender Representation of Speaking Opportunities at the American Geophysical Union Fall Meeting”

than the demographic options provided. AGU does not plan to collect more detailed gender information at this time, but this may be an option in the future.

3. The paper points out that the database does not include information on the gender or career stage of the co-conveners. Is there a plan to begin to collect this information? If not, why not? Given the importance of this information and the possible influence of all-male conveners on gender representation, more discussion of this point is necessary.

When the data was accessed in March 2016, the demographic information for the co-conveners was not available. AGU is currently in the process of importing all of the abstract and membership data so that co-convener demographic information will be available in the future, but it is still not available at this time. Additionally, some of the demographic information may not be available for co-authors and co-conveners as AGU membership is not required for these roles.

The following text was **revised** in the manuscript (Lines 73-78): **The abstract first author and primary convener must be AGU members; however, this constraint does not apply to invited authors, co-author(s), and co-convener(s). Most invited authors are AGU members. The abstract database does not currently include gender and career stage information for co-author(s) and co-convener(s). Therefore, we do not test for possible co-author(s) and co-convener(s) influence on gender parity.**

4. Several questions could be added to this analysis and would be of great interest to the field:
a. Are women authors more likely to submit abstracts to sessions with a women primary convener?

Unfortunately, this dataset cannot resolve this question. When an author submits an abstract to a proposed session, the genders of the conveners are not specified. An author may assume the gender of the conveners. After the abstract submission process, many sessions are often combined. If there was an initial author preference for a specific convener gender this is obfuscated during this shuffle and in the final abstract database. Perhaps a survey of abstract author preferences could address this question.

b. How common are all male (or all female) sessions? (Note: I attended all male session during the Fall 2017 AGU meeting, which made me want the answer to this question. The one I attended had 8/8 speakers in the oral session and 4/4 male conveners.)

Unfortunately, session-specific data is not included in the dataset. We imagine that in this sort of analysis the cells would be small and therefore make the abstract authors identifiable based on their demographics. As stated above, we currently do not have the demographic data on co-conveners and are unable to evaluate all male (or all female) conveners.

c. Does the gender representation vary by AGU sub discipline? AGU organizes their sessions by discipline and the analysis shown here should be able to be broken down for each of those.

Response to Reviewer Comments on “Gender Representation of Speaking Opportunities at the American Geophysical Union Fall Meeting”

Unless the numbers get too small? If so, this should be discussed.

We did this analysis previously and found no significant relationship between the proportion of women in a sub-discipline and gender bias. We now include these analyses in the manuscript and Supplementary Materials.

The following text was **revised** in the manuscript (Lines 140-143): **Although some Section and Focus Groups within AGU have a larger proportion of women, there is no significant correlation between the proportion of women and the rate of invited abstracts and oral presentations between men and women (Supplementary Materials).**

The following text was **revised** in the Supplementary Materials and Supplementary Figure 2: **13. As the proportion of women in a Section and Focus Group increases, the overall gender bias in invited authors, oral presentation assignments and poster presentations requests will go down.**

We find no significant correlations between invited authors, oral presentation assignments, and poster presentations requests and the proportion of women in the Section and Focus Groups (Supplementary Table 8, Supplementary Materials Figure 2). Note Union is excluded from oral presentation assignments and poster presentations requests because all Union sessions are oral presentations (i.e. there are no poster sessions).

--

Reviewer #2 (Remarks to the Author):

Review for: Gender Representation of Speaking Opportunities at the American Geophysical Union Fall Meeting

Overall, this paper has great data on a timely, relevant, and important topic. The authors have done a good job putting together all this data. The major claim of the paper is that women are not invited as often as men to speak at the AGU Fall Meeting, a result that will be of interest to others in the field, and also to the broader STEM community. These results tie in to the larger picture of factors that hinder women’s advancement in STEM fields. My suggestions/comments are:

1. Rather than merely present the data as the authors have done, they should also offer a broader narrative, i.e. give the paper a voice, so to speak. They do touch on some of the other studies that explore gender bias, but in its current form this paper does not flow very smoothly with respect to a compelling narrative. For example, rather than simply mention that their study is consistent with the finding of a leak in the pipeline, the authors could say a little more about the causes of such a leak, especially within the context of the bigger picture, and how their finding (i.e. that women are not invited as often as men to make presentations at AGU) is one piece of a larger problem of gender bias in academia. That way, this paper becomes more relevant to a broader audience than just geoscientists who attend AGU.

Response to Reviewer Comments on “Gender Representation of Speaking Opportunities at the American Geophysical Union Fall Meeting”

During revision, we focused on crafting this narrative as suggested. We revised the text throughout to pull in research from other areas. Thank you for this suggestion, as we believe it improves the accessibility of the research.

2. Explain the statistics and figures a little bit more – not everyone is familiar with what a chi-square statistic represents. And the figures could also do with a bit more explanation – just a couple of sentences should do.

The following text was revised in the manuscript (Lines 89-91): **The chi-squared test (χ^2) is used throughout to determine whether there is significant difference between the expected and observed frequencies.**

The following text was revised in the Supplementary Materials: We used **chi-squared test (χ^2)** to test the *hypotheses* numerated below. **χ^2 is used throughout to determine whether there is significant difference between the expected and observed frequencies. Symbols are μ (mean), σ (standard deviation) and n (number of individuals). Results are reported as: χ^2 (degrees of freedom, sample size) = the χ^2 value, and the associated p -value.**

The figure captions in the manuscript now read:

1. American Geophysical Union Fall Meeting Gender Demographics. Proportion of total abstracts by career stage (A) **shows male authors are a large portion of submitted abstracts.** Proportion of abstracts by gender by career stage (B) **shows female authors are concentrated in the student and early career stages.**

2. Author submissions to American Geophysical Fall Meeting by gender and career stage. Proportion of invited authors (A) **shows overall men are invited at a higher rate than women; however, when controlling for career stage, early career and mid-career women are invited at a higher rate than male colleagues.** Authors assigned oral presentations (B) **shows overall men are assigned oral presentations at a higher rate than women, but when controlling for career stage there is no significant difference.** Authors opting for posters (C) **shows women opt for poster presentations more often than men, both overall and at equivalent career stages.** Totals shown here are the proportion of total abstracts.

3. Primary convener allocations for American Geophysical Fall Meeting. Proportion of women across career stages invited by primary convener gender (A) **shows men primary conveners invite fewer women and the proportion of women invited by primary conveners' gender and career stage (B) shows this is true across men primary conveners' career stages.** Proportion of women across career stages assigned oral presentations by primary convener gender (C) **shows men primary conveners assign oral presentations to fewer women and the proportion of women assigned oral presentations by primary conveners' gender and career stage (D) shows this is true across men primary conveners' career stages.**

Response to Reviewer Comments on “Gender Representation of Speaking Opportunities at the American Geophysical Union Fall Meeting”

3. Refer to a few additional studies that are relevant to this paper – for example the Hebl paper that just came out in PNAS (Dec 2017) – that shows that women were much less likely to be invited to be colloquium speakers compared to men. <http://www.pnas.org/content/early/2017/12/12/1708414115>

The following text was **revised** in the manuscript (Lines 152-155): **The presence of women on speaker selection committees improved parity in virology conferences(7) and colloquium speakers at top universities(16). AGU should encourage women to act as primary conveners. Women co-conveners may also help improve parity.**

4. Explicitly offer a specific set of recommendations instead of merely hinting at what could be done. For example, lines 133-136 say that men primary conveners control a larger portion of abstracts, so their preference for inviting men over women has a disproportionate impact on the visibility of women speakers. Following this, make a recommendation that AGU committees should make it a point to include more women primary conveners, and/or that male primary conveners should seek input from others (including women peers and colleagues) before issuing invitations to speakers. One could also make a case that following the data and findings in this paper, AGU should require all primary conveners to go through a bias training, and also explain why they invited the speakers that they did, and what steps if any they took to ensure broader participation to include women speakers. The authors can further buttress this point by referring to the recent PlosOne paper that shows that in the natural sciences women are more egalitarian than men (see: <https://www.ncbi.nlm.nih.gov/pmc/articles/PMC5425184/?platform=hootsuite>), and so putting more women in primary convener positions would broaden participation.

The following text was **revised** in the manuscript (Lines 193-198): Encouraging more women to act as primary conveners and colloquium organizers may also reduce the overall gender imbalance(7, 16), **so AGU and other scientific conference organizers should promote women in these roles.** All conveners may benefit from implicit bias training prior to inviting and assigning oral presentations to speakers, **although additional research is needed on training efficacy(20, 21).**

5. This paper has implications for shaping policy on inviting speakers for presentations – not just AGU but potentially other STEM professional societies as well. The authors should develop this aspect further instead of merely stating the data.

The following text was **revised** in the manuscript (Lines 202-207): **We recommend AGU and similar scientific organizations include statements asking primary conveners and conference organizers to be mindful of diversity. Mindfulness about diversity issues, particularly among men, while making decisions on speaking opportunities, reviewer suggestions, job candidates, and scientific recognition in medal and awards will improve community diversity.**

Response to Reviewer Comments on “Gender Representation of Speaking Opportunities at the American Geophysical Union Fall Meeting”

6. Tie in bias (implicit or otherwise) with the leaks in the pipeline in STEM fields in academia. Be it letters of recommendation, journal reviewer activity, teaching evaluations, salary differences, grant funding, etc. – these are all pieces of a larger picture, one in which bias, combined with society’s perceptions of gender-normative roles, plays a very key role in limiting women’s professional advancement in STEM fields.

The following text was **revised** in the manuscript (Lines 177-181): **There are myriad reasons why women may leave academic STEM fields; implicit biases—from the quality of recommendation letters(3), funding(19) speaking opportunities(7, 16), and recognition and medals(6)—and explicit biases, such as structural biases that impede family formation and parental support(8), all hinder the progression of women into more senior roles and their research visibility.**

7. The ending seems very abrupt.

We hope the text included in Reviewer Comment #5 is less of an abrupt ending.

8. Overall, this paper has great data. My main concerns, as evident from my points above, are that: a) the authors have not fully developed the potential policy implications of this data; and b) the paper lacks the “connective tissue” in the form of a compelling narrative that will influence the thinking in this field and make these results accessible and relevant to a broader community. However, these limitations can be fixed with some re-writing, and do not require any major changes.

We thank the reviewer for their useful suggestions.

Reviewers' Comments:

Reviewer #1:

Remarks to the Author:

The revisions have greatly improved the paper. The only remaining comment I have is that it would be nice to have some sort of "Future work" section. In my initial comments I wanted to see a lot more data reduction, which cannot be done at the moment based on limitations in the data set. It would be helpful for the authors to put some of their comments to me into the paper in this future work section. For example, in their response to my third comment, they state "AGU is currently in the process of importing all of the abstract and membership data so that co-convener demographic information will be available in the future, but it is still not available at this time." This information would be useful for the readers of the paper, as well. It would also be good to end with some suggestions to the AGU about what data they could collect to enable more analyses of this type.

Reviewer #2:

Remarks to the Author:

This looks fine to me now, as the authors have incorporated the suggestions. Important topic and a well-written paper.

Response to Reviewer Comments on “Gender Inequity in Speaking Opportunities at a Major Scientific Conference”

Note: Reviewer comments are in normal text, *responses to reviewers are in blue italic text*, and **manuscript revisions are in bold text**.

We thank the two reviewers for their beneficial feedback.

Reviewer #1 (Remarks to the Author):

The revisions have greatly improved the paper. The only remaining comment I have is that it would be nice to have some sort of "Future work" section. In my initial comments I wanted to see a lot more data reduction, which cannot be done at the moment based on limitations in the data set. It would be helpful for the authors to put some of their comments to me into the paper in this future work section. For example, in their response to my third comment, they state "AGU is currently in the process of importing all of the abstract and membership data so that co-convener demographic information will be available in the future, but it is still not available at this time." This information would be useful for the readers of the paper, as well. It would also be good to end with some suggestions to the AGU about what data they could collect to enable more analyses of this type.

Thank you for your suggestion.

The following text was **revised** in the manuscript (Lines 196-203): **The AGU membership and abstract database is evolving into a rich resource to explore diversity and equity. AGU started collecting demographic data, including gender and ethnicity (members affiliated with institutions in the United States of America) from its membership in 2013. As response rates improve, opportunities will increase for nuanced demographic analyses. It is particularly important to understand underrepresented minority groups because geoscience is one of the least ethnically diverse STEM fields²⁰. AGU is in the process of importing all of the abstract and membership data so that co-convener demographic information will also be available for future analyses.**

Reviewer #2 (Remarks to the Author):

This looks fine to me now, as the authors have incorporated the suggestions. Important topic and a well-written paper.

Thank you again for your helpful feedback.